# Dehydroepiandrosterone Sulfate, an Adrenal Androgen, Is Inversely Associated with Prevalence of Dynapenia in Male Individuals with Type 2 Diabetes

**DOI:** 10.3390/metabo13111129

**Published:** 2023-11-03

**Authors:** Saya Yasui, Yousuke Kaneko, Hiroki Yamagami, Minae Hosoki, Taiki Hori, Akihiro Tani, Tomoyo Hara, Kiyoe Kurahashi, Takeshi Harada, Shingen Nakamura, Toshiki Otoda, Tomoyuki Yuasa, Hiroyasu Mori, Akio Kuroda, Itsuro Endo, Munehide Matsuhisa, Takeshi Soeki, Ken-ichi Aihara

**Affiliations:** 1Department of Internal Medicine, Anan Medical Center, 6-1 Kawahara Takarada-cho, Anan 774-0045, Japan; sayapon626@outlook.jp (S.Y.); yousukekaneko115@gmail.com (Y.K.); yamagami.hiroki@tokushima-u.ac.jp (H.Y.); minae.energy.flow@gmail.com (M.H.); taiki4725@yahoo.co.jp (T.H.); oniru1993@gmail.com (A.T.); 2Department of Hematology, Endocrinology and Metabolism, Tokushima University Graduate School of Biomedical Sciences, 3-18-15 Kuramoto-cho, Tokushima 770-8503, Japan; hara.tomoyo@tokushima-u.ac.jp (T.H.); kurahashi.kiyoe@tokushima-u.ac.jp (K.K.); takeshi_harada@tokushima-u.ac.jp (T.H.); 3Department of Community Medicine and Medical Science, Tokushima University Graduate School of Biomedical Sciences, 3-18-15 Kuramoto-cho, Tokushima 770-8503, Japan; shingen@tokushima-u.ac.jp (S.N.); otoda.toshiki@tokushima-u.ac.jp (T.O.); yuasa.tomoyuki@tokushima-u.ac.jp (T.Y.); soeki@tokushima-u.ac.jp (T.S.); 4Diabetes Therapeutics and Research Center, Institute of Advanced Medical Sciences, Tokushima University, 3-18-15 Kuramoto-cho, Tokushima 770-8503, Japan; mori.hiroyasu@tokushima-u.ac.jp (H.M.); kurodaakio@tokushima-u.ac.jp (A.K.); matuhisa@tokushima-u.ac.jp (M.M.); 5Department of Bioregulatory Sciences, Tokushima University Graduate School of Biomedical Sciences, 3-18-15 Kuramoto-cho, Tokushima 770-8503, Japan; endoits@tokushima-u.ac.jp

**Keywords:** dehydroepiandrosterone sulfate, presarcopenia, sarcopenia, dynapenia, type 2 diabetes

## Abstract

Dehydroepiandrosterone sulfate (DHEAS) is thought to be associated with life expectancy and anti-aging. Although skeletal muscle disorders are often found in diabetic people, the clinical significance of DHEAS in skeletal muscle remains unclear. Therefore, we aimed to determine whether DHEAS is associated with the development of skeletal muscle disorders in individuals with type 2 diabetes (T2D). A cross-sectional study was conducted in 361 individuals with T2D. Serum DHEAS levels, skeletal muscle mass index (SMI), handgrip strength (HS), and gait speed (GS) were measured in the participants. Pre-sarcopenia, sarcopenia, and dynapenia were defined according to the definitions of the AWGS 2019 criteria. DHEAS level was positively associated with HS but not with SMI or GS after adjustment of confounding factors. Multiple logistic regression analyses in total subjects showed that DHEAS level had an inverse association with the prevalence of dynapenia but not with the prevalence of pre-sarcopenia or sarcopenia. Furthermore, a significant association between DHEAS level and dynapenia was found in males but not in females. ROC curve analysis indicated that cutoff values of serum DHEAS for risk of dynapenia in males was 92.0 μg/dL. Therefore, in male individuals with T2D who have low serum levels of DHEAS, adequate exercise might be needed to prevent dynapenia.

## 1. Introduction

Populations in developed countries are aging, and significant attention has therefore been paid to geriatric syndrome. Aging-related muscle diseases, known as pre-sarcopenia, sarcopenia, and dynapenia, are included in this syndrome. Sarcopenia is defined as muscle loss and weakness, dynapenia is defined as impairment of muscle strength without reduction in muscle mass, and pre-sarcopenia is defined as reduction in muscle mass without impairment of muscle strength [1]. Previous studies have shown that these aging-related muscle diseases are more frequent in individuals with diabetes than in individuals without diabetes and that they might contribute to impaired quality of life and incidental falls [2,3]. In addition, diabatic peoples with sarcopenia have a higher risk of mortality after hospital discharge [4]. Therefore, prevention and/or prediction of skeletal muscle disorders in individuals with diabetes are pivotal clinical issues.

Dehydroepiandrosterone (DHEA) and its sulfate “DHEAS” are hormones produced by the adrenal cortex. They are mostly produced in young adulthood and their concentrations gradually decline over time [5]. Previous studies showed that a high level of DHEAS is associated with a lower risk of worsening frailty in the elderly [6] and that a lower level of DHEAS may indicate a poorer prognosis in patients with atherosclerosis [7] and cardiovascular diseases [8].

Although Yanagita et al. showed that serum cortisol-to-DHEAS ratio is strongly associated with sarcopenia in individuals with type 2 diabetes (T2D) [9], the relationships between serum DHEAS level and skeletal muscle disorders, including pre-sarcopenia, sarcopenia and dynapenia, have remained unclear. Therefore, the aim of the present study was to determine the pathophysiological role of DHEAS in skeletal muscle disorders in individuals with T2D.

## 2. Materials and Methods

### 2.1. Subjects for Cross-Sectional Study

A total of 361 Japanese subjects (202 males and 159 females) who were outpatients or inpatients with T2D and 20 years of age or older were recruited consecutively from the Department of Internal Medicine at Anan Medical Center, Anan Tokushima, Japan, between April 2020 and March 2022. All subjects underwent a standardized interview and physical examination. The diagnosis of T2D was based on the criteria proposed by the Expert Committee on the Diagnosis and Classification of Diabetes Mellitus [10]. Body mass index was calculated as an index of obesity. Blood pressure was measured twice and averaged. Hypertensive patients were defined as those with systolic blood pressure (SBP) ≥ 140 mmHg and/or diastolic blood pressure (DBP) ≥ 90 mmHg or those receiving antihypertensive agents. Patients with dyslipidemia were defined as those with low-density lipoprotein cholesterol (LDL-C) ≥ 140 mg/dL or triglycerides (TG) level ≥ 150 mg/dL or high-density lipoprotein cholesterol (HDL-C) less than 40 mg/dl or those receiving lipid-lowering agents. Current smokers were defined as subjects who had smoked within the past two years. Habitual exercisers were defined as those who engaged in aerobic exercise including walking (3 Mets or more) for 30 min or more at a time, at least three times a week for at least 1 year.

Body composition, muscle strength, and physical performance were evaluated in all the participants in this study in accordance with following described procedures.

The exclusion criteria for individuals with T2D were as follows: (1) patients with advanced cancer, (2) patients with secondary diabetes such as steroid-induced diabetes or pancreatic diabetes, (3) patients who were pregnant, (4) patients with advanced renal disease with a serum creatinine (Cr) level of >2.0 mg/dL, and (5) patients with liver cirrhosis and malnutrition (serum albumin (ALB) level of <3.0 g/dL).

### 2.2. Biochemical Analyses

Blood and spot urine samples were collected and used for determination of blood cell counts, plasma glucose (PG), HbA1c, and serum biochemical parameters including LDL-C, TG, HDL-C, ALB, uric acid (UA), and Cr. PG and serum levels of LDL-C, TG, HDL-C, ALB, UA, and Cr were measured by enzymatic methods using an automatic analyzing apparatus (LABOSPECT 008, Hitachi High-Tech Co., Tokyo, Japan). HbA1c was assayed by high-performance liquid chromatography using an analyzing apparatus (HLC-723 G11, Tosoh Co., Tokyo Japan). These biochemical analyzers have been widely used in Japanese hospitals. Serum level of DHEAS was determined by the chemiluminescent enzyme immunoassay (Access DHEA-S^®^, Beckman Coulter, Tokyo, Japan). The coefficients of variance for intra- and inter-assays in DHEAS measurements were 3.04% and 3.97%, respectively. These values mean statistically sufficient reproducibility in this examination.

### 2.3. Assessments of Muscle Strength and Physical Performance

Handgrip strength (HS) was evaluated as an indicator of muscle strength, and gait speed (GS) was determined to evaluate physical performance. The maximum isometric grip strength in each hand was measured in a standing position (103S TARZAN^®^; HATAS, Osaka, Japan). GS in each participant was measured as the time needed to walk 6 m expressed in meters per second.

### 2.4. Definitions of Robust, Presarcopenia, Sarcopenia and Dynapenia

Body composition was measured by using multifrequency bioelectrical impedance analysis as previously described in [11]. In brief, body composition analysis was performed using a portable direct segmental multifrequency bioimpedance analysis (DSM-BIA) device (InBody S10^®^, InBody Japan, Tokyo, Japan). DSM-BIA was conducted using an 8-point tactile electrode system with 30 impedance examinations taken by using 6 frequencies (1, 5, 50, 250, 500, 1000 kHz) at each of the 5 segments (bilateral thumbs, third fingers, and ankles). Patient characteristics, including information on age, sex, body weight, and height, were entered into the DSM-BIA device. Electrical currents of 1, 5, 50, 250, 500, and 1000 kHz were applied through the electrodes in the standing position. Body fat mass and skeletal muscle mass index (SMI) were calculated using formulas in the inner software based on the height, weight, and impedance examined [11].

Sarcopenia was defined as a low SMI with low HS or slow GS and pre-sarcopenia was defined as a low SMI with normal HS and normal GS according to the definitions of the EWGSOP 2019 criteria [12]. Dynapenia was defined as a normal SMI with low HS or slow GS according to the definitions of the AWGS 2019 criteria [13]. Subjects without sarcopenia, pre-sarcopenia or dynapenia were defined as robust subjects. Definitions of robust, pre-sarcopenia, sarcopenia, and dynapenia in the present study are shown in Figure 1.

### 2.5. Statistical Analysis

Normally distributed continuous data were presented as means ± standard deviation (SD). Skewed continuous data were presented as medians and interquartile range (IQR). Categorical variables were compared by performing the χ^2^ test or Fisher’s exact test. For comparisons among groups, we performed the ANOVA or Kruskal–Wallis’s test for numeric variables depending on the variables’ distribution. Simple linear regression analyses to evaluate associations of levels of DHEAS with SMI, HS, and GS were performed and multiple linear regression analyses for determinants of SMI, HS, and GS were conducted. In addition, the degrees of associations between the prevalence of each skeletal muscle disorder (pre-sarcopenia, sarcopenia, and dynapenia) and clinical variables including DHEAS were determined by performing logistic regression analysis. Receiver operating characteristic (ROC) curve analysis was conducted to determine the optimal cutoff values of serum DHEAS levels in relation to the prevalences of skeletal muscle disorders. The optimal cutoff point selection of serum DHEAS levels in the context of ROC curve analysis was determined on the basis of the maximum value of the Youden index.

The analyses were performed by using GraphPad Prism 9 (GraphPad Software, San Diego, CA, USA) and EZR (Saitama Medical Center, Jichi Medical University). The threshold for statistical significance was set at *p* < 0.05.

## 3. Results

### 3.1. Clinical Characteristics of Subjects Enrolled in This Study

The physical and laboratory-determined characteristics of subjects enrolled in this study are shown in Table 1. On average, the robust subjects were younger than the subjects with skeletal muscle disorders. The BMI values of subjects with pre-sarcopenia and sarcopenia were lower than those of robust subjects and subjects with dynapenia. Although there was no significant difference in basic clinical factors including SBP, serum lipids, ALB, PG, HbA1c, UA, and Cr among the four groups, serum level of DHEAS in the dynapenia group was significantly lower than that in the robust group. The percentage of subjects with dynapenia who did habitual exercise was lower than the percentage of robust subjects who did habitual exercise. The prevalence of dyslipidemia in subjects with pre-sarcopenia was lower than that in robust subjects. The durations of T2D in the three skeletal muscle disorder groups were longer than the duration of T2D in the robust group. The SMI values of subjects with pre-sarcopenia and sarcopenia were significantly lower than the SMI values of robust subjects and subjects with dynapenia. The HS values of subjects with sarcopenia and dynapenia were lower than the HS values of robust subjects and subjects with pre-sarcopenia. The GS values of subjects with sarcopenia and dynapenia were slower than the GS values of robust subjects and subjects with pre-sarcopenia. The percentages of subjects with sarcopenia and dynapenia using antiplatelets were higher than those of robust subjects and subjects with pre-sarcopenia.

### 3.2. Associations of Serum Levels of DHEAS with SMI, HS and GS

Simple linear regression analyses showed that serum levels of DHEAS had positive associations with all skeletal muscle-associated indices, including SMI, HS, and GS, in total subjects and male subjects with T2D (Table 2). On the other hand, simple linear regression analyses showed that serum levels of DHEAS had a positive association with HS but not with SMI or GS in female subjects with T2D (Table 2).

### 3.3. Associations of Clinical Factors Including DHEAS with SMI, HS and GS

To determine independent clinical factors associated with SMI, HS, and GS in patients with T2D, we performed multiple linear regression analyses with clinical confounding factors including DHEAS. As shown in Table 3, we found that male gender, BMI, exercise, and Cr were significantly and positively correlated with SMI. In contrast, age, SBP, and LDL-C were significantly and negatively correlated with SMI. Next, we found that male gender, BMI, exercise, Cr, ALB, and DHEAS were significantly and positively correlated with HS. Conversely, age, duration of T2D, and UA were significantly and negatively correlated with HS. In regard to determinants of GS, we observed that exercise and ALB were significantly and positively correlated with GS. On the other hand, male gender, age, and BMI were significantly and negatively correlated with GS. To assess the influence of medication used in this study on associations of identified clinical factors with SMI, HS, and GS, we performed additional multiple linear regression analyses with identified clinical factors and medications used in the participants. As shown in Appendix A, DHEAS remained a positive contributor for HS.

### 3.4. Determination of Clinical Factors for Prevalences of Presarcopenia, Sarcopenia and Dynapenia

To determine clinical factors related to the prevalences of pre-sarcopenia, sarcopenia, and dynapenia in individuals with T2D, we performed multiple logistic regression analyses in each skeletal muscle disorder with reference to robust subjects. As shown in Table 4, in terms of risk factors for prevalence of the pre-sarcopenia, male gender, lower BMI, and duration of T2D were corelated with pre-sarcopenia. In the analysis for prevalence of sarcopenia, aging and lower BMI were found to be significant risk factors. As for the prevalence of dynapenia, aging and lower DHEAS were identified as significant risk factors for the prevalence of dynapenia in individuals with T2D. Lastly, we divided the subjects into male subjects and female subjects for the identification of dynapenia-associated factors, and a significant association between DHEAS and dynapenia was found in males but not in females (Table 5). The association between DHEAS and dynapenia in males remained to be significant in a multiple logistic regression analysis, including nutritional status represented by geriatric nutritional risk index: GNRI (14.89 × ALB (g/dL) + 41.7 × body weight (kg)/ideal body weight (kg)) (Appendix A). The increase in the odds ratio of 0.985 at 1 μg/dL of DHEAS for prevalence of dynapenia in male individuals with T2D was considered that the increase in the odds ratio of 0.860 at 10 μg/dL of DHEAS for prevalence of dynapenia in male individuals with T2D.

### 3.5. Determination of Cutoff Value of Serum DHEAS Levels for Prevalence of Dynapenia in Male Individuals with T2D

We conducted ROC curve analysis to determine the optimal cutoff value of serum DHEAS levels in relation to the prevalence of dynapenia in the male subjects of the present study. The analysis showed that the optimal cutoff value was 92.0 μg/dL with the area under the curve (AUC): 0.789 (95% CI: 0.705 to 0.873) (Figure 2). This AUC value is reasonable for indicating the significance of DHEAS in identification of dynapenia in male individuals with T2D.

## 4. Discussion

In this study, we found that serum levels of DHEAS have a negative association with the prevalence of dynapenia in males with T2D but not in females with T2D.

Age-related declines of skeletal muscle mass and function are critical problems that have gained attention in clinical practice. Recent studies have shown that skeletal muscle strength deficits with aging are more rapid than the concomitant loss of skeletal muscle quantity [14]. Chao et al. reported that 18.49%, 3.52%, and 1.06% of 568 robust elderly subjects with a mean age of 70.1 years transited to dynapenia, pre-sarcopenia, and sarcopenia, respectively, during a 6-year follow up period [15]. Those results are consistent with our results, showing that the incidence of dynapenia was remarkably higher than the incidences of pre-sarcopenia and sarcopenia. The results suggest that elderly individuals have more problems with skeletal muscle strength rather than with skeletal muscle mass loss during the aging process.

It has been shown that dynapenia had a direct effect on increased mortality in 610 subjects aged 65 years or older [16]. In addition, a 10-year follow-up study involving 5310 older adults showed that the coexistence of anemia and dynapenia increased the mortality risk in both males and females [17]. Therefore, risk detection and prediction of dynapenia are urgent challenges for the promotion of health and longevity in elderly individuals with diabetes.

Although the detailed mechanisms of muscle strength weakness have not been fully clarified, aging-induced reactive oxygen species (ROS) has been proposed to impair muscle quality at various locations [14]. Muscle regeneration following damage diminishes with aging and it has been clarified that muscle regeneration is dependent on appropriate reinnervation [18]. Since accumulation of oxidative damage in nerves and muscles occurs with aging, abnormal activities of ROS promote defective muscle reinnervation throughout life [19]. Oxidative stress induces glycoxidation reactions, which lead to the formation of highly reactive and electrophilic compounds that results in the generation of advanced glycation end products (AGEs) [20,21]. Mori et al. showed that accumulated levels of AGEs represented by skin autofluorescence (SAF) were significantly higher in patients with sarcopenia and patients with dynapenia among individuals with T2D [22], and Kato et al. reported that SAF was an independent factor associated with low SMI among middle-aged and older Japanese men and women [23], suggesting that accumulation of AGEs promotes reduction of skeletal muscle mass and strength, leading to sarcopenia and/or dynapenia in elderly individuals.

It has been reported that DHEA treatment induced a marked decrease in the plasma concentration of pentosidine as a biomarker of AGEs in individuals with T2D [24] and it has been shown that DHEA treatment counteracted the enhanced AGE receptor activation in the heart of streptozotocin-diabetic rats and Zucker diabetic fatty rats and resulted in amelioration of diabetic cardiomyopathy [25]. Taken together, those results indicate the possibility that DHEA and/or DHEAS contribute to the reduction in oxidative stress accompanied by accumulation of AGEs in skeletal muscle, leading to prevention of the development of muscle weakness.

In a systematic review assessment of previous clinical studies regarding the association of endogenous serum DHEA(S) levels and skeletal muscle disorders [26], an early cross-sectional study showed that lower quadricep maximal muscle power and lower optimal shortening velocity were associated with levels of DHEAS in elderly females but not in elderly males [27]. A previous study showed that serum DHEAS was an independent predictor of muscle strength and mass in elderly males aged 60–79 years [28]; however, other studies did not show a significant association [29,30]. Although we reported that DHEAS has sex-dependent diverse vascular protective effects against carotid atherosclerosis in individuals with cardiovascular risk factors [7], the precise differential mechanisms of the sex-dependent pathophysiological effects of DHEAS have remained unclear and the diverse associations between DHEAS and skeletal muscle disorders in males and females are issues to be further investigated.

### Limitations

The main limitation of our study is the retrospective cross-sectional nature with a relatively small sample size, which precludes conclusions regarding the temporal nature of our findings and no solid conclusions can be established. Even though we found that the levels of serum DHEAS were associated with the prevalence of dynapenia in patients with T2D, we cannot confirm a causal relationship between DHEAS and incidence of dynapenia. In addition, dietary content was not considered in our analysis. Since the protein contents in daily meals contribute to the maintenance of skeletal muscle mass, further investigation regarding dietary content is needed. Another limitation of this study is that the results cannot be extended to the general population because we enrolled only subjects with T2D. For these reasons, prospective studies are required to establish the time sequence in the relationships between serum levels of DHEAS and incidences of skeletal muscle disorders including mass and strength abnormalities in elderly individuals with or without diabetes.

## 5. Conclusions

In summary, the present study showed that serum levels of DHEAS were inversely associated with the prevalence of dynapenia in male individuals with T2D. Taken together, the results indicate that efficient exercise should be performed to preserve skeletal muscle strength for healthy aging in male individuals with T2D who have low serum levels of DHEAS.

## Figures and Tables

**Figure 1 metabolites-13-01129-f001:**
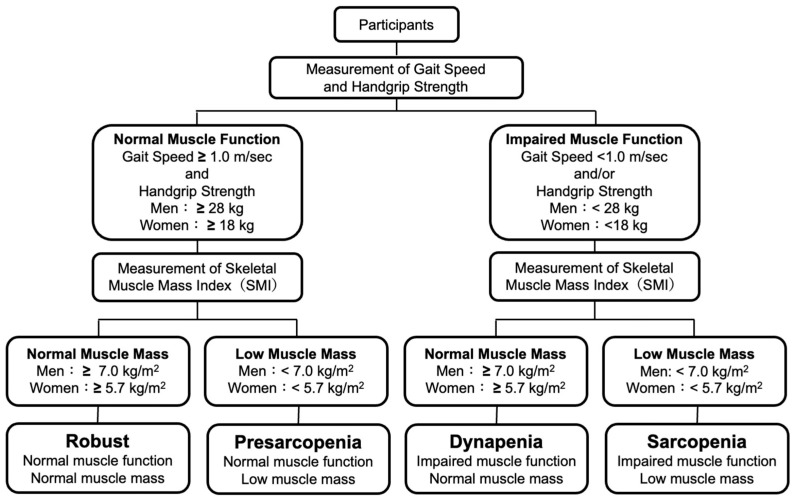
Definitions of robust and skeletal muscle disorders.

**Figure 2 metabolites-13-01129-f002:**
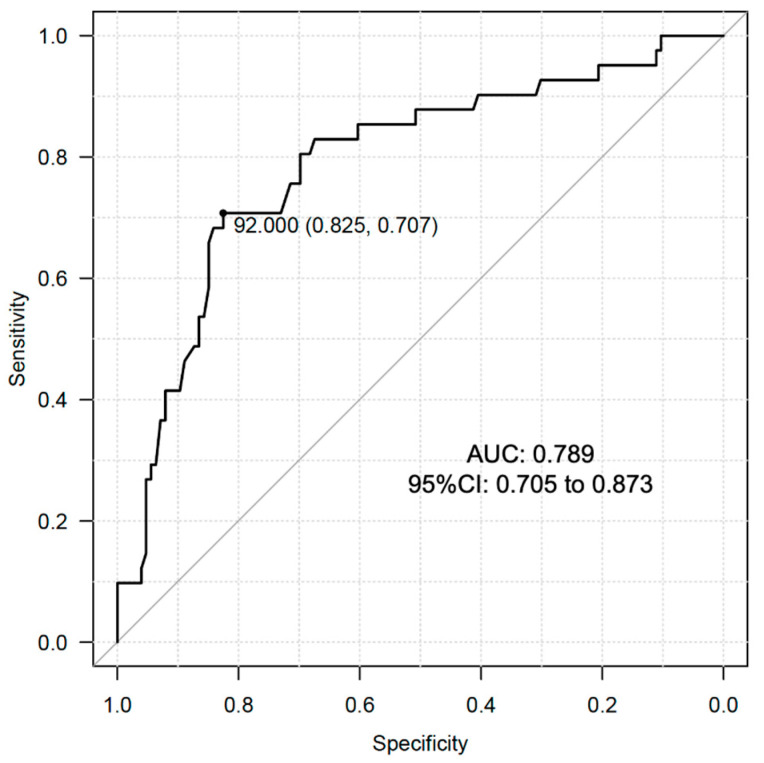
ROC curve analysis for cutoff value determination of DHEAS for the prevalence of dynapenia in male subjects with T2D.

**Table 1 metabolites-13-01129-t001:** Clinical characteristics of subjects in total, robust, pre-sarcopenia, sarcopenia, and dynapenia group.

	Total	Robust	Pre-Sarcopenia	Sarcopenia	Dynapenia
Number of Subjects	361	233	22	37	69
Clinical Parameters					
Male (Female)	202 (159)	126 (107)	16 (6)	19 (18)	41 (28)
Age (year)	69 (61, 75)	64 (58, 71)	73 (70, 78) *	76 (72, 81) *	75 (71, 79) **
BMI (kg/m^2^)	24.2 (21.9, 26.9)	25.2 (22.9, 28.0)	20.1 (19.0, 20.7) **	20.1 (18.8, 22.7) **	24.3 (22.5, 25.9) ^#$^
Exercise (≥3 Mets) (n, (%))	118 (32.7)	79 (33.9)	11 (50.0)	8 (21.6)	20 (29.0) *
Current Smoking (n, (%))	58 (16.1)	43 (18.5)	4 (18.2)	4 (10.8)	7 (10.1)
Hypertension (n, (%))	229 (63.4)	141 (60.5)	15 (68.2)	27 (75.0)	46 (66.7)
Dyslipidemia (n, (%))	274 (75.9)	183 (78.5)	10 (45.5) **	28 (75.7)	53 (76.8)
Duration of T2D (year)	9 (2, 17)	6 (1, 12)	16 (11, 27) **	12 (7, 20) *	12 (6, 20) **
SBP (mmHg)	132 (121, 143)	132 (121, 142)	127 (119, 140)	132 (121, 144)	133 (121, 143)
LDL-C (mg/dL)	98 (80, 119)	101 (81, 121)	102 (88, 116)	94 (81, 110)	89 (75, 116)
HDL-C (mg/dL)	52 (44, 62)	51 (44, 62)	61 (52, 69.8)	52 (44, 61)	53 (45, 59)
TG (mg/dL)	110 (78, 162)	121 (81, 175)	86 (69, 116)	98 (72, 131)	101 (72, 152)
Caual PG (mg/dL)	131 (113, 171)	130 (112, 162)	149 (124, 189)	124 (110, 167)	132 (117, 175)
HbA1c (%)	6.7 (6.3, 7.3)	6.7 (6.3, 7.3)	6.8 (6.5, 7.5)	6.8 (6.3, 7.2)	6.7 (6.3, 7.2)
UA (mg/dL)	4.9 (4.1, 5.9)	5.0 (4.1, 5.9)	5.2 (4.6, 5.9)	4.5 (3.8, 5.3)	4.8 (4.2, 6.0)
Cr (mg/dL)	0.74 (0.62. 0.92)	0.73 (0.61. 0.90)	0.78 (0.66. 0.96)	0.70 (0.59, 0.92)	0.77 (0.62, 1.05)
ALB (g/dL)	4.2 (4.0, 4.4)	4.3 (4.1, 4.5)	4.3 (4.0, 4.4)	4.1 (4.0, 4.3)	4.2 (3.8, 4.4)
DHEAS (μg/dL)	97 (60, 159)	107 (63, 180)	114 (76, 181)	84 (47, 124)	78 (53, 113) *
Skeletal Muscle-associated Indices					
SMI (kg/m^2^)	7.1 (6.4, 7.8)	7.4 (6.8, 8)	6.6 (5.8, 6.7) **	5.7 (5.3, 6.5) **^#^	7.1 (6.6, 7.6) ^#$$^
HS (kg)	26.5 (20.5, 33.8)	29.5 (23.0, 36.5)	27.5 (24.5, 28.4)	17.5 (14.5, 23.3) **^#^	21.8 (16.8, 26.5) **
GS (m/s)	1.22 ± 0.26	1.32 ± 0.21	1.30 ± 0.20	1.05 ± 0.26 **^#^	0.95 ± 0.19 **^#^
Medications Used					
ARB/ACEi (n, (%))	143 (39.6)	92 (39.5)	8 (36.4)	14 (37.8)	29 (42.0)
CCB (n, (%))	133 (36.8)	81 (34.5)	7 (31.8)	15 (40.5)	30 (43.5)
β blocker (n, (%))	10 (2.8)	5 (2.1)	2 (9.1)	1 (2.7)	2 (2.9)
MR blocker (n, (%))	4 (1.1)	3 (1.3)	1 (4.5)	0 (0)	0 (0)^#^
Stain (n, (%))	184 (51.0)	117 (50.2)	9 (40.9)	22 (59.5)	36 (52.2)
Ezetimibe (n, (%))	25 (6.9)	13 (5.6)	1 (4.5)	2 (5.4)	9 (13.0)
Other lipid-lowering drugs (n, (%))	7 (1.9)	4 (1.7)	0 (0)	0 (0)	3 (4.3)
Antiplatelet (n, (%))	36 (10.0)	10 (4.3)	0 (0)	11 (29.7) **^#^	15 (21.7) **^#$^
SU or Glinide (n, (%))	68 (18.8)	32 (13.7)	7 (31.8)	9 (24.3)	20 (29.0)
DPP-4i (n, (%))	207 (57.3)	124 (53.2)	12 (54.5)	27 (73.0)	44 (63.8)
Metformin (n, (%))	180 (49.9)	116 (49.8)	8 (36.4)	20 (54.1)	36 (52.2)
αGI (n, (%))	47 (13.0)	24 (10.3)	2 (9.1)	7 (19.0)	14 (20.3)
Glitazone (n, (%))	11 (3.0)	5 (2.1)	2 (9.1)	1 (2.7)	3 (4.3)
SGLT2i (n, (%))	147 (40.7)	98 (42.1)	7 (31.8)	11 (29.7)	31 (44.9)
Insulin (n, (%))	71 (19.7)	40 (17.2)	8 (36.4)	9 (24.3)	14 (20.3)
GLP-1RA (n, (%))	37 (10.2)	23 (9.9)	2 (9.1)	4 (10.8)	8 (11.6)

Abbreviations: ACEi: angiotensin-converting enzyme inhibitor; ALB: albumin; ARB: angiotensin receptor blocker; αGI: alpha glucosidase inhibitor; BMI: body mass index; CCB: calcium channel blocker; Cr: serum creatinine; DBP: diastolic blood pressure; DHEAS: dehydroepiandrosterone sulfate; DPP-4i: dipeptidyl peptidase 4 inhibitor; eGFR: estimated glomerular filtration rate; GLP1-RA: glucagon-like peptide 1 receptor agonist; GS: gait speed; HbA1c hemoglobin A1c; HDL-C: high-density lipoprotein cholesterol; HS: handgrip strength; LDL-C: low-density lipoprotein cholesterol; MR: mineral corticoid receptor; PG: plasma glucose; SBP: systolic blood pressure; SGLT2i: sodium-glucose cotransporter 2 inhibitor; SMI: skeletal muscle mass index; SU: sulfonyl urea; TG: triglycerides; UA: uric acid. * *p* < 0.05 vs. robust, ** *p* < 0.01 vs. robust, # *p* < 0.01 vs. pre-sarcopenia, $ *p* < 0.05 vs. sarcopenia, $$ *p* < 0.01 vs. sarcopenia.

**Table 2 metabolites-13-01129-t002:** Simple linear regression analysis for associations of DHEAS with SMI, HS and GS in total, male and female individuals with T2D.

	Total			Males			Females		
Skeletal Muscle-Associated Indices	Coefficient	95% CI	*p* Value	Coefficient	95% CI	*p* Value	Coefficient	95% CI	*p* Value
SMI	0.004	0.003 to 0.006	<0.001	0.003	0.001 to 0.004	<0.001	0.002	−0.001 to 0.004	0.059
HS	0.046	0.036 to 0.056	<0.001	0.035	0.022 to 0.049	<0.001	0.017	0.005 to 0.030	0.006
GS	0.001	0.000 to 0.001	0.002	0.001	0.000 to 0.001	<0.001	0.001	−0.001 to 0.001	0.853

Abbreviations: CI: confidence interval; DHEAS: dehydroepiandrosterone sulfate; GS: gait speed; HS: handgrip strength; SMI: skeletal muscle mass index.

**Table 3 metabolites-13-01129-t003:** Multiple linear regression analysis for determinants of SMI, HS, and GS in total individuals with T2D.

	SMI			HS			GS		
Variables	t Value	VIF	*p* Value	t Value	VIF	*p* Value	t Value	VIF	*p* Value
Age	−8.048	1.727	<0.001	−7.213	1.727	<0.001	−5.469	1.717	<0.001
Male	14.500	1.692	<0.001	11.760	1.692	<0.001	−2.051	1.695	0.041
BMI	20.670	1.584	<0.001	2.481	1.584	0.014	−2.356	1.521	0.019
Exercise (≥3 Mets)	2.925	1.100	0.004	2.213	1.100	0.028	2.643	1.103	0.009
Current Smoking	0.980	1.237	0.328	1.167	1.237	0.244	−0.484	1.238	0.628
Hypertension	0.747	1.360	0.456	0.596	1.360	0.552	−0.078	1.362	0.938
Dyslipidemia	−1.752	1.198	0.081	−0.753	1.198	0.452	−0.662	1.200	0.509
Duration of T2D	0.512	1.228	0.609	−3.091	1.228	0.002	0.365	1.232	0.715
SBP	−2.654	1.268	0.008	−0.547	1.268	0.585	0.602	1.286	0.547
LDL-C	−2.784	1.216	0.006	−1.740	1.216	0.083	1.419	1.226	0.157
HDL-C	0.985	1.470	0.326	0.051	1.470	0.959	0.313	1.459	0.754
TG	0.096	1.442	0.923	0.039	1.442	0.969	0.593	1.480	0.554
HbA1c	0.018	1.219	0.985	−0.124	1.219	0.901	−0.968	1.223	0.334
UA	1.467	1.435	0.143	−2.289	1.435	0.023	0.618	1.422	0.537
Cr	2.880	1.420	0.004	3.248	1.420	0.001	1.153	1.407	0.250
ALB	−1.314	1.229	0.190	3.803	1.229	<0.001	2.691	1.231	0.008
DHEAS	−0.802	1.381	0.423	2.822	1.381	0.005	1.053	1.409	0.293

Abbreviations: ALB: albumin; BMI: body mass index; Cr: serum creatinine; DHEAS: dehydroepiandrosterone sulfate; GS: gait speed; HbA1c hemoglobin A1c; HDL-C: high-density lipoprotein cholesterol; HS: handgrip strength; LDL-C: low-density lipoprotein cholesterol; SBP: systolic blood pressure; SMI: skeletal muscle mass index; TG: triglycerides; UA: uric acid; VIF: variance inflation factor.

**Table 4 metabolites-13-01129-t004:** Multiple logistic regression analysis for determinants of pre-sarcopenia, sarcopenia, and dynapenia in total individuals with T2D.

	Pre-Sarcopenia				Sarcopenia				Dynapenia			
Variables	OR	95% CI	VIF	*p* Value	OR	95% CI	VIF	*p* Value	OR	95% CI	VIF	*p* Value
Age (1 year)	1.022	0.950 to 1.107	1.632	0.562	1.241	1.127 to 1.403	1.724	<0.001	1.147	1.092 to 1.213	1.645	<0.001
Male	14.03	1.734 to 176.6	1.762	0.023	1.066	0.251 to 4.518	1.726	0.930	1.984	0.844 to 4.860	1.718	0.124
BMI (1 kg/m^2^)	0.390	0.217 to 0.582	1.520	<0.001	0.597	0.452 to 0.745	1.519	<0.001	1.053	0.961 to 1.155	1.538	0.264
Exercise (≥3 Mets)	1.076	0.240 to 4.859	1.164	0.923	0.311	0.080 to 1.047	1.097	0.072	0.498	0.236 to 1.012	1.105	0.060
Current Smoking	0.839	0.148 to 4.026	1.263	0.830	1.063	0.181 to 5.435	1.269	0.943	0.996	0.322 to 2.877	1.264	0.994
Hypertension	1.944	0.376 to 11.11	1.400	0.434	1.991	0.579 to 7.521	1.349	0.287	1.090	0.508 to 2.358	1.378	0.825
Dyslipidemia	0.547	0.110 to 2.613	1.219	0.447	2.358	0.583 to 10.86	1.162	0.245	0.879	0.379 to 2.088	1.189	0.765
Duration of T2D (1 year)	1.063	1.008 to 1.128	1.250	0.029	0.990	0.942 to 1.039	1.261	0.682	1.016	0.983 to 1.050	1.220	0.343
SBP (1 mmHg)	0.975	0.918 to 1.030	1.279	0.371	1.028	0.989 to 1.063	1.289	0.151	0.991	0.968 to 1.013	1.267	0.446
LDL-C (1 mg/dL)	0.997	0.968 to 1.024	1.262	0.849	0.994	0.974 to 1.014	1.242	0.586	0.996	0.984 to 1.008	1.247	0.479
HDL-C (1 mg/dL)	1.033	0.978 to 1.091	1.481	0.233	0.973	0.924 to 1.021	1.459	0.282	1.010	0.983 to 1.037	1.471	0.474
TG (1 mg/dL)	1.001	0.987 to 1.013	1.421	0.916	0.991	0.979 to 1.003	1.423	0.145	1.002	0.996 to 1.007	1.425	0.526
HbA1c (1%)	1.099	0.709 to 1.577	1.201	0.623	1.261	0.894 to 1.744	1.212	0.163	1.011	0.798 to 1.243	1.234	0.922
UA (1 mg/dL)	1.595	0.850 to 3.126	1.451	0.153	1.261	0.783 to 2.067	1.452	0.345	1.159	0.861 to 1.565	1.480	0.331
Cr (1 mg/dL)	0.781	0.007 to 14.52	1.375	0.915	0.138	0.006 to 2.418	1.362	0.193	0.471	0.081 to 1.859	1.441	0.380
ALB (1 g/dL)	1.516	0.166 to 11.88	1.236	0.698	0.959	0.139 to 6.965	1.259	0.966	0.496	0.199 to 1.250	1.218	0.130
DHEAS (1 μg/dL)	0.999	0.989 to 1.009	1.513	0.884	0.997	0.987 to 1.006	1.498	0.556	0.993	0.987 to 0.998	1.396	0.014

Abbreviations: ALB: albumin; BMI: body mass index; CI: confidence interval; Cr: serum creatinine; DHEAS: dehydroepiandrosterone sulfate; GS: gait speed; HbA1c hemoglobin A1c; HDL-C: high-density lipoprotein cholesterol; HS: handgrip strength; LDL-C: low-density lipoprotein cholesterol; OR: odds ratio; SBP: systolic blood pressure; SMI: skeletal muscle mass index; TG: triglycerides; UA: uric acid; VIF: variance inflation factor.

**Table 5 metabolites-13-01129-t005:** Multiple logistic regression analysis for determinants of dynapenia in male and female individuals with T2D.

	Dynapenia							
	Males				Females			
Variables	OR	95% CI	VIF	*p* Value	OR	95% CI	VIF	*p* Value
Age (1 year)	1.142	1.060 to 1.247	2.114	0.001	1.261	1.147 to 1.418	1.528	<0.001
BMI (1 kg/m^2^)	1.021	0.875 to 1.193	1.883	0.792	0.987	0.857 to 1.134	1.357	0.849
Exercise (≥3 Mets)	0.307	0.104 to 0.833	1.108	0.025	1.028	0.266 to 3.729	1.166	0.967
Current Smoking	1.244	0.301 to 4.991	1.295	0.757	4.608	0.159 to 61.720	1.097	0.280
Hypertension	1.449	0.472 to 4.495	1.561	0.516	1.048	0.268 to 4.130	1.467	0.945
Dyslipidemia	0.321	0.082 to 1.179	1.277	0.092	2.481	0.520 to 15.060	1.162	0.245
Duration of T2D (1 year)	1.056	0.998 to 1.122	1.287	0.066	0.986	0.930 to 1.039	1.307	0.597
SBP (1 mmHg)	0.974	0.940 to 1.008	1.472	0.139	1.014	0.982 to 1.040	1.337	0.297
LDL-C (1 mg/dL)	0.997	0.979 to 1.016	1.332	0.751	0.992	0.971 to 1.013	1.273	0.457
HDL-C (1 mg/dL)	1.005	0.961 to 1.049	1.380	0.807	0.996	0.950 to 1.043	1.459	0.282
TG (1 mg/dL)	1.005	0.998 to 1.014	1.462	0.199	0.995	0.984 to 1.006	1.548	0.370
HbA1c (1%)	0.847	0.547 to 1.180	1.346	0.369	1.048	0.688 to 1.492	1.258	0.812
UA (1 mg/dL)	1.444	0.899 to 2.429	1.313	0.144	0.816	0.452 to 1.429	1.714	0.485
Cr (1 mg/dL)	0.666	0.060 to 2.382	1.228	0.674	0.053	0.001 to 1.512	1.607	0.102
ALB (1 g/dL)	0.362	0.091 to 1.248	1.332	0.119	0.834	0.123 to 5.885	1.274	0.852
DHEAS (1 μg/dL)	0.985	0.975 to 0.993	1.366	0.001	1.008	0.999 to 1.016	1.298	0.059

Abbreviations: ALB: albumin; BMI: body mass index; CI: confidence interval; Cr: serum creatinine; DHEAS: dehydroepiandrosterone sulfate; GS: gait speed; HbA1c hemoglobin A1c; HDL-C: high-density lipoprotein cholesterol; HS: handgrip strength; LDL-C: low-density lipoprotein cholesterol; OR: odds ratio; SBP: systolic blood pressure; SMI: skeletal muscle mass index; TG: triglycerides; UA: uric acid; VIF: variance inflation factor.

## Data Availability

The datasets generated in the present study are available from the corresponding author upon reasonable request. Data is not publicly available due to privacy.

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
