# Peer review of "Dehydroepiandrosterone Sulfate, an Adrenal Androgen, Is Inversely Associated with Prevalence of Dynapenia in Male Individuals with Type 2 Diabetes"

_metabolites, 2023, doi:10.3390/metabo13111129_

Round 1
Reviewer 1 Report
Comments and Suggestions for Authors
This study investigates the relationship between DHEAS levels and skeletal muscle health in individuals with type 2 diabetes, revealing that low DHEAS levels in males with T2D may be indicative of an increased risk of dynapenia, suggesting the potential significance of exercise interventions for preventing this condition. But there are certain points that need careful attention of the authors to revise this manuscript:
1. How were the subjects for the cross-sectional study selected, and were there any specific inclusion or exclusion criteria related to their type 2 diabetes (T2D) status?
2The elaboration for the biochemical analysis of DHEAS, LDL-C, TG, HDL-C, albumin, uric acid, and creatinine is very limited. Their brief description regarding their analysis along with the details of assay kits are required.
3. Please explain the significance of using the coefficients of variance for intra- and inter-assays in DHEAS measurements.
4. What were the specific numerical findings regarding the associations between DHEAS levels and skeletal muscle disorders, such as dynapenia, in males and females with T2D?
5. How were the cutoff values for serum DHEAS levels determined in relation to the prevalences of skeletal muscle disorders in males with T2D?
Author Response
We would like to thank Reviewer 1 for the valuable comments on the original version of our manuscript. We have addressed each concern pointed out by the reviewer.
Reviewer 1
Comments and Suggestions for Authors
This study investigates the relationship between DHEAS levels and skeletal muscle health in individuals with type 2 diabetes, revealing that low DHEAS levels in males with T2D may be indicative of an increased risk of dynapenia, suggesting the potential significance of exercise interventions for preventing this condition. But there are certain points that need careful attention of the authors to revise this manuscript:
- How were the subjects for the cross-sectional study selected, and were there any specific inclusion or exclusion criteria related to their type 2 diabetes (T2D) status?
(Response)
In accordance with the reviewer’s comment, we added a detailed explanation of the exclusion criteria in the revised manuscript as follows: Body composition, muscle strength and physical performance were evaluated in all of the participants in this study in accordance with following described procedures. The exclusion criteria for individuals with T2D were as follows: 1) patients with advanced cancer, 2) patients with secondary diabetes such as steroid-induced diabetes or pancreatic diabetes, 3) patients who were pregnant, 4) patients with advanced renal disease with a serum creatinine (Cr) level of >2.0 mg/dL and 5) patients with liver cirrhosis and malnutrition (serum albumin (ALB) level of < 3.0 g/dL).
- The elaboration for the biochemical analysis of DHEAS, LDL-C, TG, HDL-C, albumin, uric acid, and creatinine is very limited. Their brief description regarding their analysis along with the details of assay kits are required.
(Response)
In accordance with the reviewer’s comment, we added an explanation of the biochemical analysis in the revised manuscript as follows: PG and serum levels of LDL-C, TG, HDL-C, ALB, UA and Cr were measured by enzymatic methods using an automatic analyzing apparatus (LABOSPECT 008, Hitachi High-Tech Co., Tokyo, Japan). HbA1c was assayed by high-performance liquid chromatography using an analyzing apparatus (HLC-723 G11, Tosoh Co., Tokyo Japan). These biochemical analyzers have been widely used in Japanese hospitals.
- Please explain the significance of using the coefficients of variance for intra- and inter-assays in DHEAS measurements.
(Response)
The statement in our manuscript that “The coefficients of variance for intra- and inter-assays in DHEAS measurements were 3.04% and 3.97%, respectively” indicated very small variance ranges of less than 5% and these values mean statistically sufficient reproducibility in this examination.
- What were the specific numerical findings regarding the associations between DHEAS levels and skeletal muscle disorders, such as dynapenia, in males and females with T2D?
(Response)
As shown in Table 5 and Figure 2 of the revised manuscript, multiple regression analysis and ROC curve analysis showed sufficient significance of DHEAS for the prevalence of dynapenia in male subjects. On the other hand, Table 5 shows no significant association between DHEAS and prevalence of dynapenia in female subjects. In addition, ROC curve analysis failed to show diagnostic capacity of DHEAS for prevalence of dynapenia in female subjects (AUC: 0.44; 95% CI: 0.325 to 0.555).
In addition, we added the following sentence in the revised manuscript: The increase in the odds ratio of 0.985 at 1 μg/dL of DHEAS for prevalence of dynapenia in male individuals with T2D was considered that the increase in the odds ratio of 0.860 at 10 μg/dL of DHEAS for prevalence of dynapenia in male individuals with T2D.
- How were the cutoff values for serum DHEAS levels determined in relation to the prevalences of skeletal muscle disorders in males with T2D?
(Response)
We added a description about the issue in the revised manuscript as follows: And the optimal cutoff point selection of serum DHEAS levels in the context of ROC curve analysis was determined on the basis of the maximum value of the Youden index.
Reviewer 2 Report
Comments and Suggestions for Authors
Saya Yasui and collaborators have carried out a clinical study that concludes that, in men with type 2 diabetes mellitus, there is a clear relationship between serum levels of dehydroepiandrosterone sulfate and the risk of developing dynapenia. Overall, from my point of view, the work is well designed and carried out. The manuscript, likewise, is well written and described.
However, I think there are some minor aspects that still need to be improved:
- Point 2.2. It is very important to indicate how all the indicated parameters were measured (what device was used, what technique, etc.) or, at least, include a bibliographic reference detailing the methods.
- Line 85. Likewise, include a bibliographic citation describing the latex agglutination method.
- Line 120. It is described that the ROC curve was used to calculate the optimal cut-off point...based on what was this cut-off point selected? maximum sensitivity? maximum specificity? Youden index? this should be explained.
- Figure 2. I think that performing a linear regression with this data is not the best option (seeing the distribution of the points in the figures), and inserting the regression line in the figure does not seem very appropriate to me either. Personally, I think it would be more appropriate to only carry out a correlation analysis between both variables and not try to determine if it fits a straight line (since clearly, it can be seen that the points are not distributed in the form of a straight line).
- Tables 3 and 4. All the abbreviations that appear in them need to be included in the table footer.
- Line 199. I think that a cut-off point is only a value, above which and below which two types of populations are distributed. I think that using the less than or equal symbol is not correct when indicating a cut-off point, the cut-off point is only = 92 μg/dL.
- Figure 3. I consider that an important part of the analysis of the ROC curve is missing... could you include the area under the curve and statistically verify if this ROC curve has significant diagnostic capacity?
Author Response
We would like to thank Reviewer 2 for the valuable suggestions on the original version of our manuscript. We have addressed each concern pointed out by the reviewer.
Reviewer 2
Comments and Suggestions for Authors
Saya Yasui and collaborators have carried out a clinical study that concludes that, in men with type 2 diabetes mellitus, there is a clear relationship between serum levels of dehydroepiandrosterone sulfate and the risk of developing dynapenia. Overall, from my point of view, the work is well designed and carried out. The manuscript, likewise, is well written and described.
However, I think there are some minor aspects that still need to be improved:
- Point 2.2. It is very important to indicate how all the indicated parameters were measured (what device was used, what technique, etc.) or, at least, include a bibliographic reference detailing the methods.
(Response)
In accordance with the reviewer’s comment, we added an explanation of biochemical analysis in the revised manuscript as follows:
PG and serum levels of LDL-C, TG, HDL-C, ALB, UA and Cr were measured by enzymatic methods using an automatic analyzing apparatus (LABOSPECT 008, Hitachi High-Tech Co., Tokyo, Japan). HbA1c was assayed by high-performance liquid chromatography using an analyzing apparatus (HLC-723 G11, Tosoh Co., Tokyo Japan). These biochemical analyzers have been widely used in Japanese hospitals.
In brief, body composition analysis was performed using a portable direct segmental multifrequency bioimpedance analysis (DSM-BIA) device (InBody S10®️, InBody Japan, Tokyo, Japan). DSM-BIA was conducted using an 8-point tactile electrode system with 30 impedance examinations taken by using 6 frequencies (1, 5, 50, 250, 500, 1000 kHz) at each of the 5 segments (bilateral thumbs, third fingers and ankles). Patient characteristics including information on age, sex, body weight, and height were entered into the DSM-BIA device. Electrical currents of 1, 5, 50, 250, 500, and 1000 kHz were applied through the electrodes in the standing position. Body fat mass and skeletal muscle mass index (SMI) were calculated using formulas in the inner software based on the height, weight, and impedance examined11.
- Line 85. Likewise, include a bibliographic citation describing the latex agglutination method.
(Response)
Since latex agglutination for HbA1c measurement was wrong, I corrected the description about the measurement of HbA1c in the revised manuscript as shown in the above response.
- Line 120. It is described that the ROC curve was used to calculate the optimal cut-off point...based on what was this cut-off point selected? maximum sensitivity? maximum specificity? Youden index? this should be explained.
(Response)
The same concern was raised by Reviewer 1. In accordance with the reviewer’s comment, we added a description about the issue in the revised manuscript as follows: And the optimal cutoff point selection of serum DHEAS levels in the context of ROC curve analysis was determined on the basis of the maximum value of the Youden index.
- Figure 2. I think that performing a linear regression with this data is not the best option (seeing the distribution of the points in the figures), and inserting the regression line in the figure does not seem very appropriate to me either. Personally, I think it would be more appropriate to only carry out a correlation analysis between both variables and not try to determine if it fits a straight line (since clearly, it can be seen that the points are not distributed in the form of a straight line).
(Response)
In accordance with the reviewer’s comment, we replaced Figure 2 (scatter plots) in the original version with new Table 2 indicating simple regression analyses of associations between DHEAS levels and the skeletal muscle-associated indices in total, male and female subjects in the revised manuscript and described as follows: Simple linear regression analyses showed that serum levels of DHEAS had positive associations with all skeletal muscle-associated indices, including SMI, HS and GS, in total subjects and male subjects with T2D (Table 2). On the other hand, simple linear regression analyses showed that serum levels of DHEAS had a positive association with HS but not with SMI or GS in female subjects with T2D (Table 2).
- Tables 3 and 4. All the abbreviations that appear in them need to be included in the table footer.
(Response)
In accordance with the reviewer’s comment, we added the abbreviations in each table footer of the revised manuscript.
- Line 199. I think that a cut-off point is only a value, above which and below which two types of populations are distributed. I think that using the less than or equal symbol is not correct when indicating a cut-off point, the cut-off point is only = 92 μg/dL.
(Response)
In accordance with the reviewer’s comment, we corrected the cutoff value of serum DHEAS level as 92 μg/dL in the revised manuscript.
- Figure 3. I consider that an important part of the analysis of the ROC curve is missing... could you include the area under the curve and statistically verify if this ROC curve has significant diagnostic capacity?
(Response)
In accordance with the reviewer’s comment, we added a description about the issue as follows: The analysis showed that the optimal cutoff value was 92.0 μg/dL with the area under the curve (AUC): 0.789 (95%CI: 0.705 to 0.873) (Figure 2). This AUC value is reasonable for indicating the significance of DHEAS in identification of dynapenia in male individuals with T2D.